# Analysis of Difference in Areal Density Aluminum Equivalent Method in Ionizing Total Dose Shielding Analysis of Semiconductor Devices

Mingyu Liu [1,2,3], Chengfa He [1,2,*], Jie Feng [1,2,*], Mingzhu Xun [1,2], Jing Sun [1,2], Yudong Li [1,2] and Qi Guo [1,2]

[1] Xinjiang Technical Institute of Physics and Chemistry, Chinese Academy of Sciences, Urumqi 830011, China; liumingyu21@mails.ucas.ac.cn (M.L.)
[2] Xinjiang Key Laboratory of Electronic Information Material and Device, Urumqi 830011, China
[3] University of Chinese Academy of Sciences, Beijing 100049, China
[*] Correspondence: hecf@ms.xjb.ac.cn (C.H.); fengjie@ms.xjb.ac.cn (J.F.)

**Abstract:** The space radiation environment has a radiation effect on electronic devices, especially the total ionizing dose effect, which seriously affects the service life of spacecraft on-orbit electronic devices and electronic equipment. Therefore, it is particularly important to enhance the radiation resistance of electronic devices. At present, many scientific research institutions still use the areal density equivalent aluminum method to calculate the shielding dose. This paper sets five common metal materials in aerospace through the GEANT4 Monte-Carlo simulation tool MULASSIS, individually calculating the absorption dose caused by single-energy electrons and protons in the silicon detector after shielding of five different materials, which have the same areal density of 0.8097 g/cm$^2$. By comparing the above data, it was found that depending on the particle energy, the areal density aluminum equivalent method would overestimate or underestimate the absorbed dose in the shielded silicon detector, especially for the ionization total dose shielding effect of low-energy electrons. The areal density aluminum equivalent method will greatly overestimate the shielding dose, so this difference needs to be taken into account when evaluating the ionizing dose of the electronics on a spacecraft to make the assessment more accurate.

**Keywords:** Monte-Carlo method; total ionizing dose; radiation shielding; space radiation

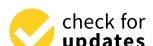



## 1. Introduction

When a spacecraft is in orbit, it inevitably experiences the influence of the space environment, including high-energy electrons, protons, and other heavy ions from the Earth's radiation belts, solar cosmic rays, and galactic cosmic rays. These high-energy particles can significantly impact the performance of semiconductor devices or circuit systems. When these high-energy particles interact with sensitive regions of the devices, they can cause a Total Ionizing Dose effect (TID), Displacement Damage Dose effect (DDD), and Single Event Effect (SEE) [1]. As a result, semiconductor devices may degrade or fail, leading to the potential paralysis of the entire electronic system. Space radiation effects are one of the critical factors contributing to the failure of spacecraft electronic devices and circuit systems, which severely affects the spacecraft's operational life in orbit. This effect is particularly noteworthy, as semiconductor device feature sizes are becoming increasingly smaller, reaching the nanometer scale, and demanding particular attention to the impact of space radiation on semiconductor devices.

Adding shielding to integrated circuits or devices in sensitive areas of spacecraft can effectively mitigate the impact of high-energy particles in space. This passive protection method is widely used, and aluminum is the most common material used due to its excellent metallic properties. The most prevalent shielding method for payload (instruments or equipment carried by satellites or spacecraft) protection is 3 mm aluminum shielding [2]. However,

different materials exhibit significant variations in shielding effectiveness under the same thickness due to their density, atomic number (Z), electron density, and other factors. For instance, high atomic number (Z) metals can effectively shield against ionization effects caused by space electrons, but they can also lead to stronger bremsstrahlung [3–6]. On the other hand, for protons, lower atomic number materials provide better shielding effectiveness [7–10]. This is because materials with lower atomic numbers have higher electron density, resulting in greater energy deposition of protons in the material and, consequently, reducing the energy of protons reaching the sensitive layer of semiconductor devices.

In order to accurately assess the potential dose levels that sensitive areas of spacecraft may be exposed to, on-orbit dose simulation is particularly crucial. Currently, commonly used radiation dose simulation methods both domestically and internationally include SHIELDOSE-2 [11] and the Monte-Carlo method [12]. Developed by the National Institute of Standards and Technology (NIST) in the United States, SHIELDOSE-2 can calculate the depth–dose relationship in spacecraft aluminum shielding materials based on electron and proton spectra. Presently, research institutions or entities worldwide predominantly employ this program to calculate shielding doses for spacecraft. However, SHIELDOSE-2 can only compute dose distribution in aluminum material shielding, leading researchers and engineers to primarily employ the areal density aluminum equivalent method. This method transforms other materials into aluminum material thickness using equal areal densities. In this scenario, the dose after aluminum shielding at this thickness is considered to be the dose after shielding with the respective material. For instance, the Space Systems Analysis Tool (SSAT) [13] developed by the European Space Agency (ESA) divides the full solid angle of payload-sensitive regions into several small sectors, traces the ray path through materials using straight lines, and then converts to the geometric thickness, which will be accumulated, of aluminum material based on the areal density aluminum equivalent method. Finally, the dose is calculated based on the depth–dose distribution of particles in the aluminum material. Another approach is using full Monte-Carlo simulations to compute shielding doses of three-dimensional models for the payload [14]. However, this method is computationally intensive, time-consuming, and less commonly used and still in the developmental stage. Common Monte-Carlo simulation software includes EGS*, MNCP**, NOVICE***, Geant4****, etc. (*http://rcwww.kek.jp/research/egs/; **https://mcnp.lanl.gov/; ***http://www.empc.com/novice.php; ****http://geant4.cern.ch/) For instance, the Geant4-based Monte-Carlo simulation tool MULASSIS (The version of MULASSIS used in this article is v1.26) [15] (multi-layered shielding simulation software) can calculate the actual ionizing dose after shielding with any material. Therefore, this study investigates the differences between the areal density aluminum equivalent method and the material Monte-Carlo simulation method in dose calculation. This article utilizes MULASSIS version 1.26, which is built upon the Geant4 toolkit version 4.10.1p3. MULASSIS is an application developed based on the Geant4 toolkit, and it automatically selects the appropriate physics processes. For electrons, it uses the "em_opt3" physics process, while for protons, it employs the "QBBC" physics process. MULASSIS exhibits a statistical error of less than 1% in ionization dosimetry calculations, which is within an acceptable range. Utilizing the Geant4-based Monte-Carlo simulation tool MULASSIS, the study conducts simulation research on the dose of single-energy electrons and protons under the same mass thickness shielding of different shielding materials. It is found that the areal density aluminum equivalent method may overestimate or underestimate ionizing dose for device exposure, depending on the particle energy. This research provides a theoretical basis for payload shielding design optimization.

## 2. Materials and Methods

MULASSIS is a one-dimensional, multi-layered radiation shielding simulation program based on the Geant4 Monte-Carlo transport software, developed through collaboration between QinetiQ, BiRA, and ESA. It allows for the simulation and calculation of the shielding effectiveness and flux analysis of various shielding materials against space

radiation environments. Users can establish models by defining parameters such as particle sources, different shielding materials, and their respective thicknesses.

In this study, the commonly used 3 mm Al equivalent shielding thickness for spacecraft was set as the material constraint. Six materials most commonly used in satellite payloads (aluminum, lead, tantalum, tungsten, molybdenum, and titanium) were chosen as validation targets. The simulation was conducted to calculate the absorbed dose in a silicon detector under equivalent areal density shielding conditions. The material information is presented in Table 1 below.

**Table 1.** Shielding material information [16].

| Material | Atomic Number (Z) | Density (g/cm$^3$) | Equivalent Areal Density (g/cm$^2$) | Geometric Thickness (mm) | Electron Density (10$^{23}$ e/g) |
|---|---|---|---|---|---|
| Aluminum | 13 | 2.699 | 0.8097 | 3.000 | 2.901 |
| Titanium | 22 | 4.540 | 0.8097 | 1.783 | 2.719 |
| Molybdenum | 42 | 10.220 | 0.8097 | 0.792 | 2.636 |
| Tantalum | 73 | 16.654 | 0.8097 | 0.486 | 2.429 |
| Tungsten | 74 | 19.300 | 0.8097 | 0.420 | 2.424 |
| Lead | 82 | 11.350 | 0.8097 | 0.713 | 2.383 |

The simulation model established for this study is a planar slab model. The first layer is set as the shielding material with a thickness of the equivalent 3 mm aluminum areal density (0.8097 g/cm$^2$), using different materials. The second layer represents the absorbing body of the silicon detector, and the third layer is a 5 mm thick layer of aluminum. This setup is designed to simulate the potential backscattering effects from materials such as circuit boards, instrument bases, and outer shells located beneath the device during on-orbit satellite operation. For a 2 MeV electron, the impact of backscattering on the absorbed dose in a silicon detector is significant. When shielded by aluminum, the ionizing dose generated by backscattering accounts for 34.92% of the total dose. It also allows for a convenient comparison with the aluminum shielding material. Regarding the choice of the thickness for the second layer (sensitive area), the MULASSIS internal algorithm computes the average energy deposition for each layer as a whole. Therefore, it is essential to select an appropriate thickness to ensure more accurate dose calculations. As shown in Figure 1, after applying 0.8097 g/cm$^2$ mass shielding, the dose distribution for 2 MeV electrons with a 200 μm thick silicon detector is displayed. At the interface, there is a significant gradient in the dose distribution, with higher doses closer to the shielding material. Thus, a thinner silicon layer makes the detector more sensitive. Considering that the sensitive region thickness of the silicon detector is on the order of micrometers, a thickness of 20 μm is chosen to ensure the accuracy of the silicon absorption dose.

The schematic diagram of the shielding model is illustrated in Figure 2 (the entire structure needs to be modeled in MULASSIS). In this diagram, the silicon detector represents the sensitive volume for ionizing total dose damage of the electronic device, and we calculated the energy deposition at each layer. However, the absorbed dose at this location is the primary focus of the study. Taking aluminum as an example, the model begins with a 3 mm Al shielding layer, followed by a 20 μm thick silicon detector, and, finally, a 5 mm Al layer. The simulation calculates the absorbed dose in the silicon detector under these conditions. Table 2 presents the parameters of incident particles. To facilitate the comparison of differences between different materials at different energies, the energy spectrum was set as monoenergetic electrons and monoenergetic protons. The energy selection is mainly based on the fact that the energies of electrons and protons in the space environment are less than 7 MeV and 300 MeV, respectively. For low-energy electrons (energy less than 2 MeV) and protons (energy less than 25 MeV), its range in Al is less than 3 mm. Therefore, we chose this energy range.

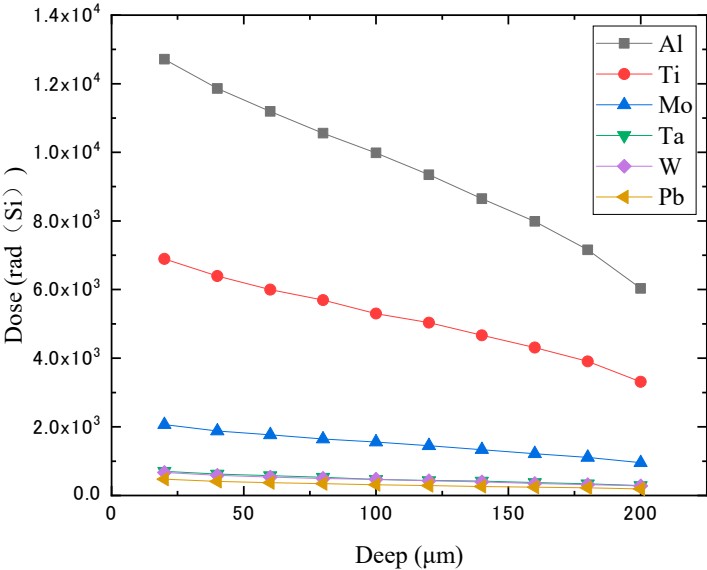

**Figure 1.** Depth–dose relationship of 2 MeV electron incidence in 200 μm silicon material.

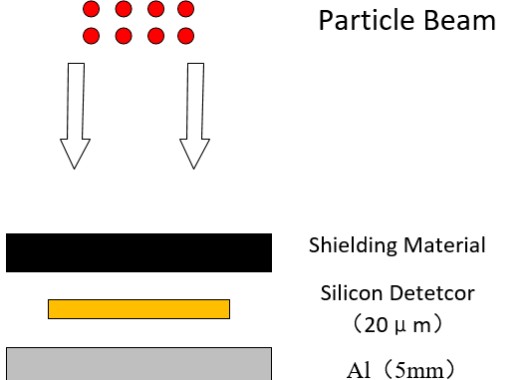

**Figure 2.** Schematic diagram of the shielding model.

**Table 2.** Particle parameters.

| Particle | Number of Primary Particles | Fluence/(Flux) Intensity $(cm^{-2}s^{-1})$ | Energy (MeV) |
|---|---|---|---|
| Electron | $1.0 \times 10^7$ | $1.0 \times 10^{12}$ cm$^{-2}$s$^{-1}$ | 2–7 |
| Proton | $1.0 \times 10^7$ | $7.6 \times 10^9$ cm$^{-2}$s$^{-1}$ | 25–300 |

## 3. Results

### 3.1. Monoenergetic Electrons

As shown in Figure 3, the relationship between absorbed dose in the silicon detector and incident electron energy was obtained for different materials under the equivalent areal density shielding (3 mm Al). A smaller absorbed dose in the silicon detector indicates a better shielding effectiveness of the material against the ionizing total dose effect caused by electrons. Overall, within the discussed range of electron energy in Figure 3, high atomic number materials such as tantalum, tungsten, and lead showed a continuous increase in ionizing dose. On the other hand, low atomic number materials such as aluminum, titanium, and molybdenum exhibited a trend of initial increase followed by a decrease in ionizing dose, with a peak point observed. For example, in the case of aluminum shielding, a distinct peak absorbed dose was observed around 4 MeV electron energy, and this peak point shifted to the right with an increase in the atomic number (Z) of the

material. Furthermore, there were significant differences in the ionizing dose between the areal density aluminum equivalent method and Monte-Carlo simulation for the same material at the same energy. When the electron energy was less than 4 MeV, the areal density aluminum equivalent method significantly overestimated the absorbed dose in the silicon detector for the other shielding materials. In Figure 4a, for a 2 MeV electron, it can be observed that aluminum shielding resulted in the highest absorbed dose in the silicon detector, while lead shielding resulted in the lowest absorbed dose, with approximately a 96% difference from aluminum. At this point, the shielding effectiveness of lead is 24.8 times that of aluminum. Indeed, for lead, the areal density aluminum equivalent method would overestimate the ionizing dose by a factor of 24.8. It would be 17.8 times for tungsten, 16.8 times for tantalum, 5.9 times for molybdenum, and 1.8 times for titanium. Until 4 MeV, the areal density aluminum equivalent method would begin to underestimate the shielding dose of other materials. In Figure 4b, for 7 MeV electron incidence, the ionizing dose after aluminum shielding is the smallest. At this point, the areal density aluminum equivalent method would underestimate the ionizing dose for tantalum, tungsten, and lead by approximately 72%, for molybdenum by 74%, and for titanium by 88%. Hence, when using the areal density aluminum equivalent method to calculate the total ionizing dose effect of electrons for other materials, there may be differences compared to Monte-Carlo simulations. These differences are closely related to the incident electron energy.

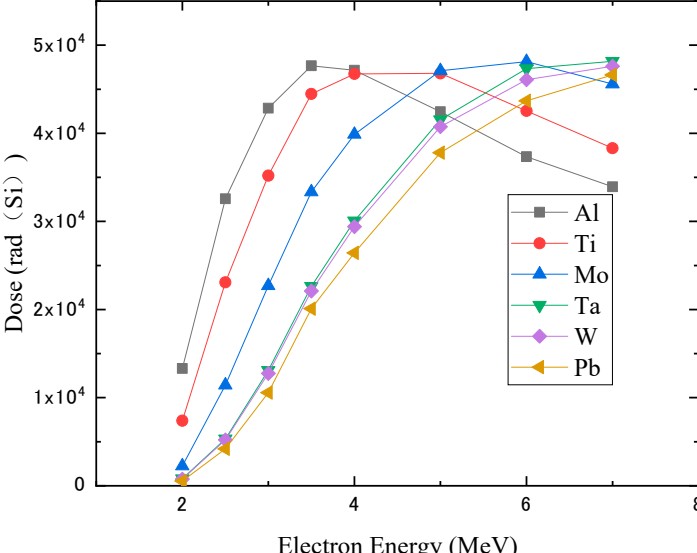

**Figure 3.** The absorbed dose in the silicon detector for six different materials (aluminum, lead, tungsten, tantalum, molybdenum, and titanium) under the same equivalent areal density shielding (3 mm Al) at various incident electron energies.

To comprehensively evaluate the shielding effectiveness of different materials with an equivalent areal density of 3 mm Al, the ionizing dose at each energy point from Figure 3 was integrated over energy. Based on the results from Table 3, when shielding with 0.8097 g/cm² areal density, the differences between the areal density aluminum equivalent method and the Monte-Carlo simulation in ionizing dose calculations was significantly reduced for a wide range of electron energy spectra. For lead, the areal density aluminum equivalent method overestimates the dose by 27.9%, for tungsten by 22.3%, for tantalum by 20.9%, for molybdenum by 7.6%, and there is almost no difference for titanium. Indeed, the areal density aluminum equivalent method for calculating ionization dose possesses certain rationality. However, there is room for further optimization and improvement.

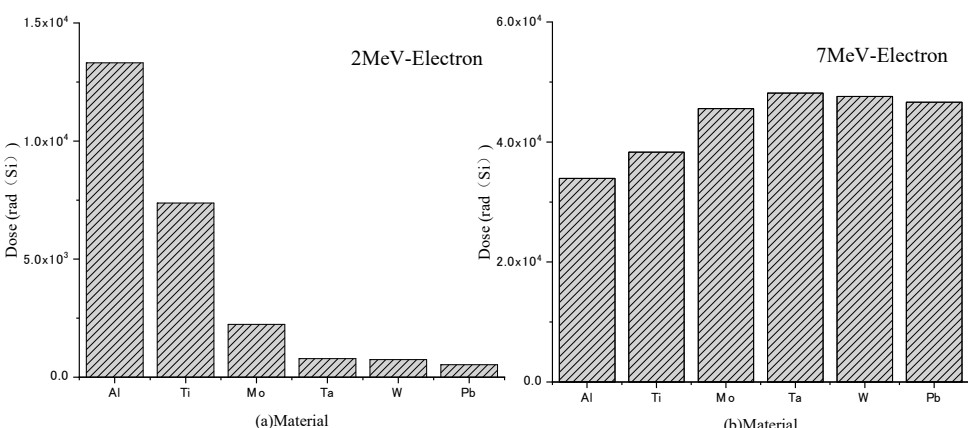

**Figure 4.** The absorbed dose in the silicon detector for six different materials (aluminum, lead, tungsten, tantalum, molybdenum, and titanium) after equivalent 3 mm Al shielding at an incident electron energy of 2 MeV (**a**) and 7 MeV (**b**).

**Table 3.** Integration of ionizing dose for 2–7 MeV electrons.

| Material | Al | Ti | Mo | Ta | W | Pb |
|---|---|---|---|---|---|---|
| Dose | $1.97 \times 10^5$ | $1.97 \times 10^5$ | $1.82 \times 10^5$ | $1.56 \times 10^5$ | $1.53 \times 10^5$ | $1.42 \times 10^5$ |
| $D_x / D_{Al}$ | 1 | 0.999 | 0.924 | 0.792 | 0.777 | 0.721 |

*3.2. Monoenergetic Protons*

Figure 5 illustrates the relationship between the absorbed dose in the silicon detector and the proton energy for six different materials after being shielded with an equivalent 3 mm Al thickness. Simultaneously, for ease of observation of the differences, we have inserted ionizing dose graphs for proton energies below 50 MeV. It can be observed that when the proton energy is less than 50 MeV, there are significant differences in the shielding effects among different materials. At this energy range, the areal density aluminum equivalent method significantly deviates from the absorbed dose obtained with actual shielding materials, leading to an overestimation of the shielding dose. The absorbed dose in the silicon detector increases as the atomic number (Z) of the materials decreases. In Figure 6, the absorbed dose in the silicon detector is shown for different materials after shielding with the same areal density at 25 MeV proton energy. Among the various materials, the absorbed dose after shielding with Ta, W, and Pb is approximately 65% higher than that after shielding with Al, while Mo shows a difference of 58%, and Ti has a difference of 43%. The absorbed dose in the silicon detector after the same surface density shielding shows an inverse relationship with the atomic number Z, and there are noticeable differences among different materials. As the proton energy increases beyond 50 MeV, the absorbed dose in the silicon detector gradually decreases and levels off at the same value for all materials. This indicates that at higher energies, the deposited dose in the sensitive area becomes independent of the atomic number of the shielding materials. Both the areal density aluminum equivalent method and the Monte-Carlo method yield similar results for calculating the shielding dose effectiveness at this energy range. Just like in electronic analysis, ionizing dose integrated over 25–300 MeV protons after shielding results in the data shown in Table 4. It can be observed that for a broad energy spectrum, the areal density aluminum equivalent method for calculating ionizing dose exhibits relatively small differences. This also indicates the method's reasonable validity.

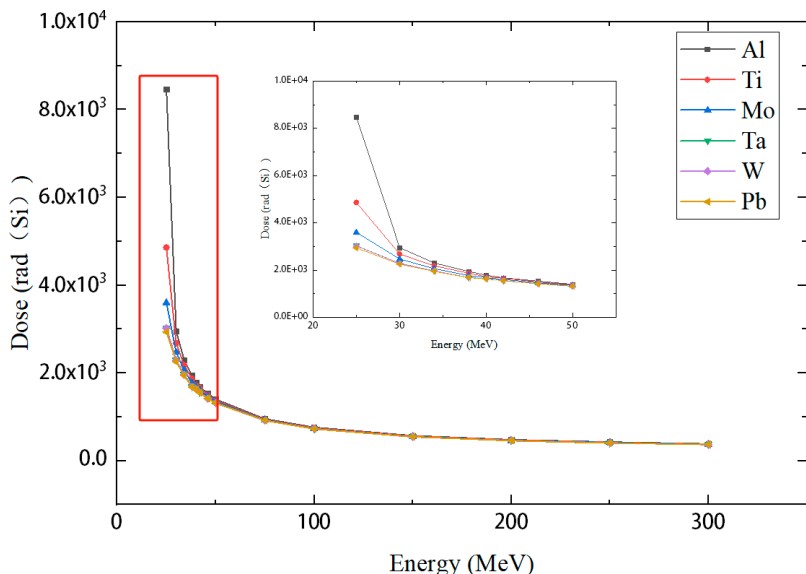

**Figure 5.** The comparison of absorbed dose in the silicon detector for different materials after single-energy proton irradiation under a shielding thickness of 0.8097 g/cm$^2$.

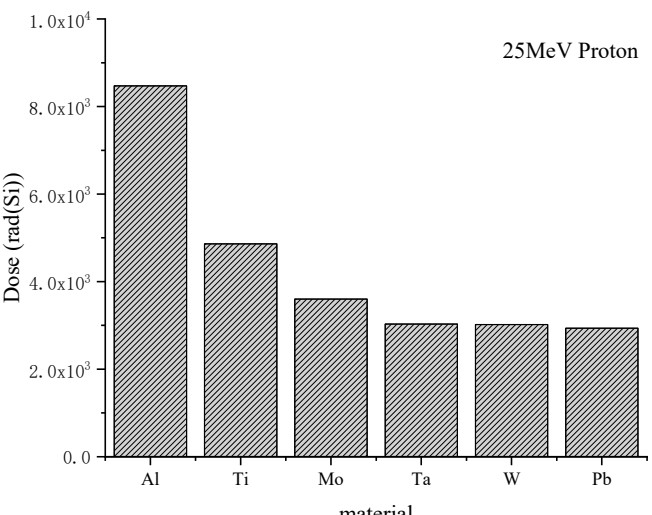

**Figure 6.** The comparison of absorbed dose in the silicon detector for different materials after shielding with a thickness of 0.8097 g/cm$^2$ under 25 MeV proton irradiation.

**Table 4.** Integration of ionizing dose for 25–300 MeV protons.

| Material | Al | Ti | Mo | Ta | W | Pb |
|---|---|---|---|---|---|---|
| Dose | $2.20 \times 10^5$ | $2.06 \times 10^5$ | $1.98 \times 10^5$ | $1.93 \times 10^5$ | $1.93 \times 10^5$ | $1.92 \times 10^5$ |
| $D_x/D_{Al}$ | 1 | 0.936 | 0.902 | 0.879 | 0.878 | 0.875 |

## 4. Discussion

### 4.1. Monoenergetic Electrons

For electrons, their energy loss within the shielding material is primarily through ionization and bremsstrahlung radiation processes [17]:

$$\left(-\frac{d_x}{d_t}\right) = \left(-\frac{d_x}{d_t}\right)_e + \left(-\frac{d_x}{d_t}\right)_r \tag{1}$$

In the above equation, the first term $\left(-\frac{d_x}{d_t}\right)_e$ represents the ionization energy loss, and the second term $\left(-\frac{d_x}{d_t}\right)_r$ represents the bremsstrahlung energy loss. The sum of these two terms gives the total energy loss of the electron. An increase in ionization energy loss within the shielding material leads to a reduction in the total dose in the silicon detector, while an increase in bremsstrahlung energy loss results in an increase in the total dose in the silicon detector. At lower energies, the contribution of bremsstrahlung is relatively small, and the absorbed dose in silicon is mainly caused by ionization of the residual electrons that penetrate the shielding material. The ionizing dose of the residual electrons in silicon can be calculated using the following formula:

$$D_e = \int_{E_0}^{E_{max}} \varnothing \left(\frac{d_E}{d_x}\right)_{Si} d_E \tag{2}$$

where $\varnothing$ represents the electron flux, and $\left(\frac{d_E}{d_x}\right)_{Si}$ denotes the collision stopping power of electrons in silicon, which characterizes the ionization energy loss of electrons in silicon. $E_0$, $E_{max}$ correspond to the minimum and maximum energy of the electrons, respectively. By utilizing the particle flux calculation function in MULASSIS, we obtained the residual electron energy spectra (Figure 7) for 2 MeV electron incidence on three different materials, Al, Mo, and Pb, after passing through an areal density of 0.8097 g/cm$^2$ shielding (as shown in Table 2, the flux of electrons before the shielding is $1.0 \times 10^{12}$ cm$^{-2}$s$^{-1}$). It is evident from Figure 7 that the remaining electron flux in Al is significantly higher than in Mo and Pb, with Pb exhibiting the lowest flux. This trend is inversely proportional to the atomic number (Z) of the materials. Consequently, at lower energies, the areal density aluminum equivalent method tends to overestimate the ionizing dose.

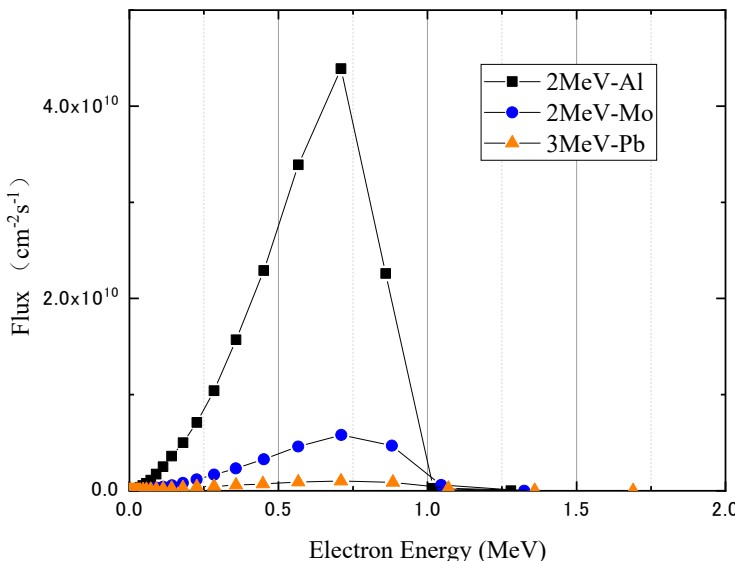

**Figure 7.** The remaining electron energy spectra after 2 MeV electron incidence on three different materials.

As the electron energy increases, the bremsstrahlung becomes more pronounced within the shielding material. In Figure 8, the radiative stopping power of 2 MeV to 7 MeV electrons in Al, Mo, and Pb materials was calculated using the ESTAR tool developed by the National Institute of Standards and Technology (NIST). It is evident that as the incident electron energy increases, the radiative stopping power also increases, with higher atomic number materials exhibiting a faster increase. The bremsstrahlung X-ray photons generated within the shielding material transfer energy to electrons in silicon through processes such as the photoelectric effect, Compton scattering, and electron–positron pair production, resulting in ionizing dose. As the radiation loss within the shielding material increases, the

absorbed dose in silicon also increases. The ionizing dose caused by X-ray photons can be calculated using the following formula:

$$D_r = \int_0^{E_{max}} \Psi \frac{\mu_{en}}{\rho} d_E, \tag{3}$$

where $\Psi$ represents the photon energy flux, $\frac{\mu_{en}}{\rho}$ denotes the mass energy absorption co-efficient of photons in Si material ($\mu_{en}$ is the coefficient of linear energy absorption, and $\rho$ is the density of the material through which the rays pass), and $D_r$ stands for dose, which represents the dose deposited in silicon by secondary bremsstrahlung produced by electrons in the shielding material. This dose can be obtained by integrating $\Psi \frac{\mu_{en}}{\rho}$ over the energy range.

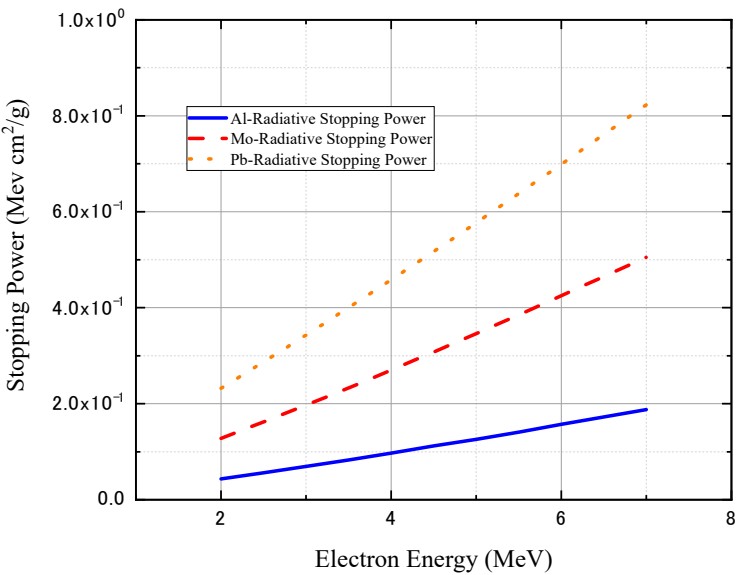

**Figure 8.** The radiative stopping power of 2 MeV to 7 MeV electrons in Al, Mo, and Pb materials.

Through the MULASSIS particle flux calculation function, with other parameters held constant, simulations were conducted for electron incident energies of 2 MeV and 7 MeV, obtaining the X-ray energy spectra after passing through Al, Mo, and Pb shielding materials, as shown in Figure 9. At both energy levels, high atomic number shielding materials produce more secondary X-rays. Integrating the energy E over the energy spectrum yields the photon energy fluence spectra (Figure 10), and integrating $\Psi \frac{\mu_{en}}{\rho}$ over the X-ray photon energy E provides the dose deposited by X-rays in Si material (Figure 11). Notably, the dose from bremsstrahlung increases by two orders of magnitude for Pb shielding material from 2 MeV to 7 MeV (from $3.07 \times 10^2$ rad(Si) to $1.328 \times 10^4$ rad(Si)), whereas for Al, it only increases by one order of magnitude (from $1.313 \times 10^2$ rad(Si) to $2.584 \times 10^3$ rad(Si)). The difference in bremsstrahlung dose between these two materials increases by 59.1 times. Mo shielding material lies between the two (increasing from $2.573 \times 10^2$ rad (Si) to $8.041 \times 10^3$ rad (Si)). Thus, for different metal materials, higher atomic numbers lead to a greater increase in secondary X-ray ionizing dose. This is due to the fact that cross-sections for the photoelectric effect ($\sigma_k \propto Z^5$), Compton scattering ($\sigma_c \propto Z$), and electron–positron pair production ($\sigma_p \propto Z^2$) are proportional to the atomic number Z raised to the fifth power, first power, and square, respectively [17]. Larger atomic numbers result in higher probabilities for these three energy transfer processes, leading to stronger bremsstrahlung. The subsequent decrease in total dose in the peak region may be attributed to the increase in electron velocity to a certain value, reducing the number of collisions with outer-shell electrons of target atoms in the shielding material. Additionally, the influence of the sensitive region's thickness contributes to the reduction in ionizing dose in the 20 μm Si detector.

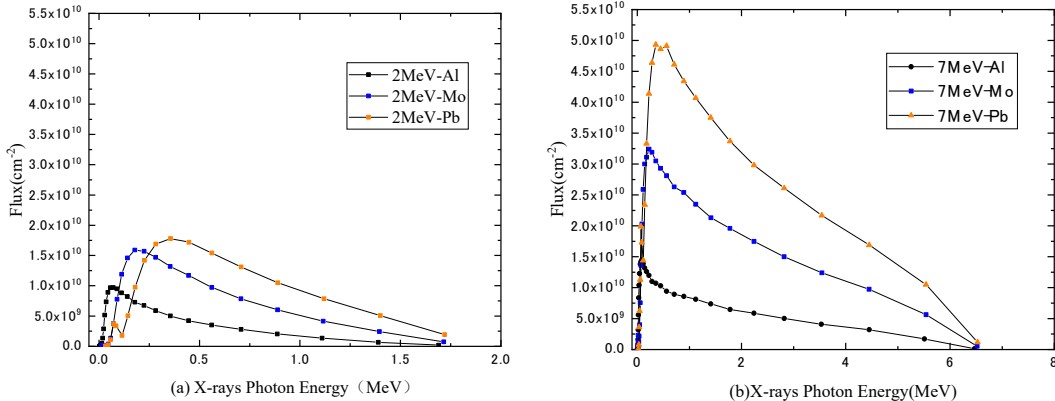

**Figure 9.** X-ray fluence spectra for 2 MeV (**a**) and 7 MeV (**b**) electrons after passing through Al, Mo, and Pb materials with the same mass shielding.

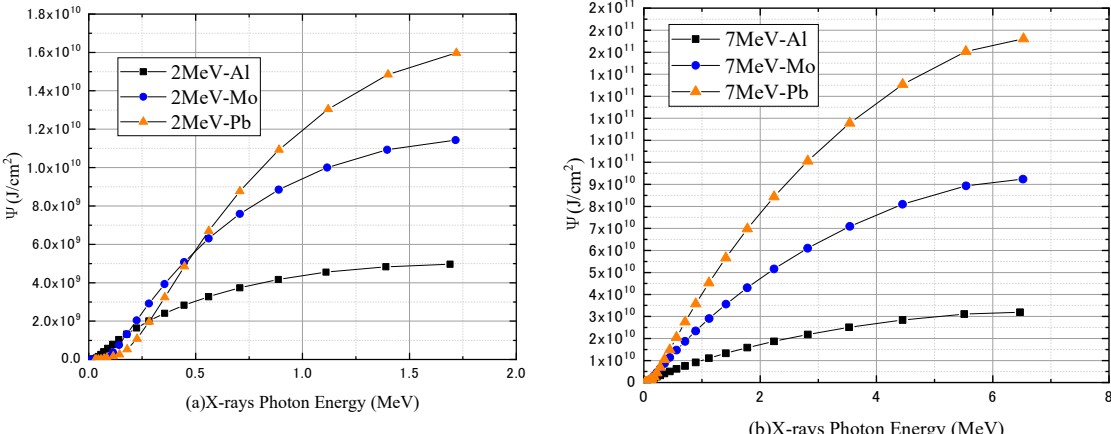

**Figure 10.** X-ray photon energy-fluence spectra for 2 MeV and 7 MeV electrons after passing through Al, Mo, and Pb materials.

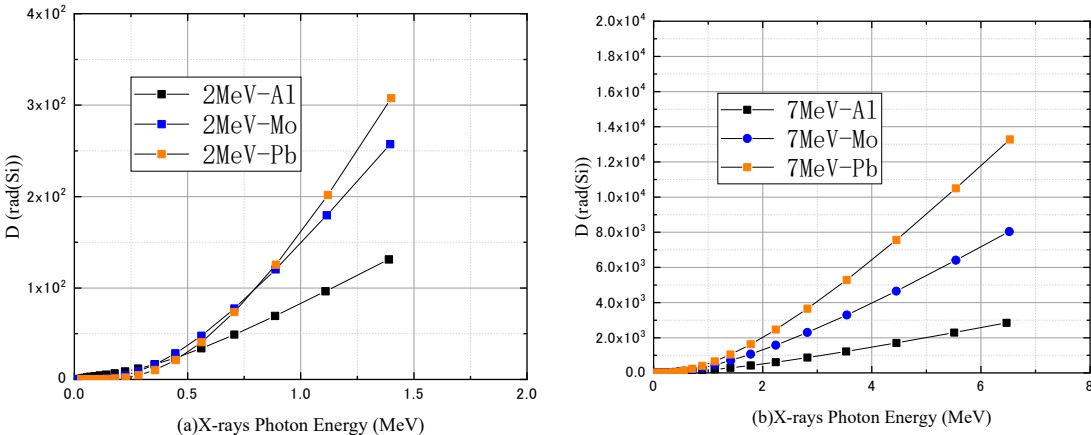

**Figure 11.** The ionizing dose produced by X-ray photons in Si material.

### 4.2. Monoenergetic Protons

For protons, their interaction with the target material involves a process of energy loss and deceleration. Upon entering the target material, protons undergo collisions with the atomic nuclei or the outer-shell electrons of the target material, leading to a continuous loss of energy and a gradual reduction in their velocity. This process continues until the proton's energy is reduced to zero, and it comes to a stop, becoming stationary within the material. The primary mechanism responsible for proton ionization effects is the energy loss resulting

from collisions between the protons and the outer-shell electrons of the target material. As protons lose energy through these collisions, they transfer energy to the electrons, leading to the ionization of atoms in the material. Figure 12 shows the depth–dose curves of 25 MeV protons in Al, Mo, and Pb metals, with the areal density used as the horizontal axis for comparison. In Figure 13, all three materials exhibit a distinct peak in the depth–dose distribution, known as the Bragg peak. The formation of the Bragg peak is due to the reduction in proton velocity as it penetrates deeper into the target material. As the proton's velocity decreases, its energy loss through collisions with the outer-shell electrons of the target material increases, leading to an increase in ionizing dose. At the end of the proton's trajectory, it comes to a stop, resulting in the maximum deposition of energy and the highest ionizing dose, forming the peak. The ionizing dose is larger on the left side of the Bragg peak and decreases with an increase in atomic number Z. For 0.8097 g/cm$^2$ Al shielding, it is closest to the Bragg peak, resulting in the highest ionizing dose in the Si material downstream of the shielding. As the proton energy increases, the 0.8097 g/cm$^2$ shielding thickness moves further away from the Bragg peak. Additionally, due to the increasing proton energy, its energy loss within the 0.8097 g/cm$^2$ thickness decreases. Figure 13 shows the depth–dose distribution after 100 MeV proton irradiation in Al, Mo, and Pb materials. At this energy, the 0.8097 g/cm$^2$ shielding for all three materials is located in the plateau region, far from the Bragg peak. Consequently, in Figure 10, with increasing energy, the dose after 3 mm equivalent Al shielding gradually decreases and eventually levels off to the same level.

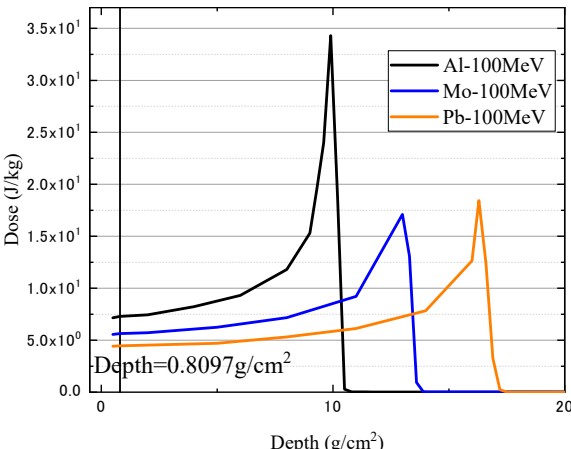

**Figure 12.** The depth–dose distribution of 25 MeV protons in three different materials.

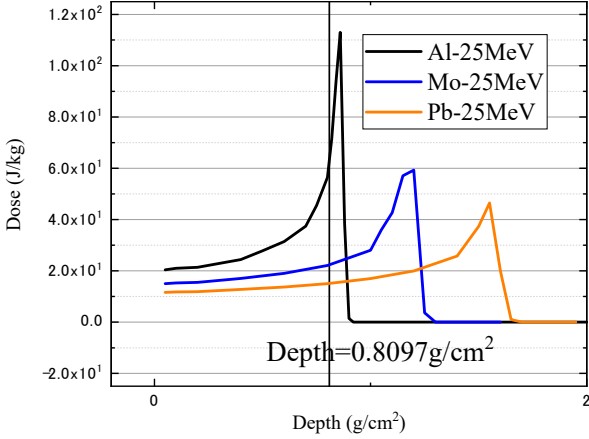

**Figure 13.** The depth–dose distribution of 100 MeV protons in three different materials.

## 5. Conclusions

In this study, we utilized the Monte-Carlo simulation tool MULASSIS in Geant4 to investigate the shielding effects on ionizing total dose in microelectronic devices for single-energy electrons or protons after shielding with lead, tungsten, tantalum, molybdenum, and titanium materials. We compared the absorbed doses and the areal density aluminum equivalent method shielding dose calculations at areal density for the materials. Our findings are as follows:

1. For single-energy electrons (2–7 MeV), significant differences exist between the areal density aluminum equivalent method and the Monte-Carlo (MULASSIS) method when calculating the absorbed dose in the silicon detector after shielding with different materials at an equivalent 3 mm Al areal density. At lower energies, the areal density aluminum equivalent method severely underestimates the shielding effectiveness of the other five materials against total ionizing dose, and the difference in absorbed dose in the silicon detector increases with larger atomic number differences. For instance, at 2 MeV electron energy, the absorbed dose in the silicon detector after lead shielding differs by 96% compared to aluminum shielding. For electron space environments with energies below 5 MeV, materials with higher atomic numbers seem to provide better shielding effects against ionization in microelectronic devices under the same areal density shielding. However, for electrons with energies above 5 MeV, the absorbed dose in the silicon detector is lower after aluminum shielding compared to other materials. Therefore, when evaluating the total dose effects caused by electrons, using the areal density aluminum equivalent method for dose assessment will overestimate the shielding effectiveness of other materials.

2. For protons, under a 0.8097 $g/cm^2$ areal density shielding, the areal density aluminum equivalent method overestimates the shielding effectiveness of the other five materials when proton energy is below 50 MeV. For example, after shielding with materials such as Pb, W, and Ta, the absorbed dose is around 35% of the dose obtained with aluminum shielding. However, as the proton energy increases, when the proton energy is greater than 50 MeV, the absorbed dose gradually converges to the same level. Therefore, for the ionizing effects caused by protons in microelectronic devices, the areal density aluminum equivalent method is not sufficiently accurate when the proton energy is below 50 MeV.

In conclusion, in radiation shielding design for payload protection, using only the areal density aluminum equivalent method to evaluate the total dose effects caused by single-energy electrons and protons may lead to inaccuracies in dose assessment. Radiation shielding and passive protection must consider the differences in dose among different materials. This approach helps accurately evaluate the radiation dose in sensitive areas of the payload and allows for targeted radiation protection designs for different regions. Strengthening the radiation resistance of devices and electronic equipment in orbit can increase the on-orbit lifetime of satellite space missions. Furthermore, in this study, single-energy electrons and protons were chosen as the subjects for comparing the differences in the ionizing total dose effects among different metal materials. However, for continuous spectrum radiation in space with various types of particles, further research is needed to investigate the variability in shielding effectiveness.

**Author Contributions:** Conceptualization, M.L. and C.H.; methodology, M.L., C.H. and J.F.; software, M.L. and M.X.; validation, M.L. and C.H.; formal analysis, C.H. and M.X.; investigation, J.F.; data curation, M.L.; writing—original draft preparation, M.L.; writing—review and editing, J.F. and C.H.; visualization, M.L.; supervision, C.H.; project administration, J.S., Y.L. and Q.G.; funding acquisition, J.F. and C.H. All authors have read and agreed to the published version of the manuscript.

**Funding:** This research was funded by the National Natural Science Foundation of China under grant No. 12175307, the National Natural Science Foundation of China under grant No. 11975305, the West Light Talent Training Plan of the Chinese Academy of Sciences under grant No. 2022-XBQNXZ-010, the West Light Talent Training Plan of the Chinese Academy of Sciences under grant No. 2021-XBQNXZ-020, and the Youth Science and Technology Talents Project of Xinjiang Uygur

Autonomous Region No. 2022TSYCCX0094, Fund of Robot Technology Used for Special Environment Key Laboratory of Sichuan Province No. 21kftk03.

**Data Availability Statement:** Not applicable.

**Conflicts of Interest:** The authors declare no conflict of interest.

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
