# Peer review of "Analysis of Difference in Areal Density Aluminum Equivalent Method in Ionizing Total Dose Shielding Analysis of Semiconductor Devices"

_electronics, doi:10.3390/electronics12194181_

Round 1
Reviewer 1 Report
This paper shows that the areal density aluminum equivalent method (ADAEM) is inadequate to predict the absorbed dose in shielded Si for high energy electrons and protons. The problems with ADAEM have been pointed our earlier for biological targets (Badavi et al, NASA/TP-2009-215779, ntrs.nasa.gov), but is here shown quantitatively for Si by comparing ADAEM to modern MC simulation tools. The paper is relatively well written but discusses the shielded dose in Si material and does not include effects in “advanced electronic devices and circuits”, which is the topic of this special issue. Despite the lack of originality and that it is slightly off the aim for this issue, it can have some interest, but there are several points that needs to be clarified (major revision). They are listed below, roughly in order of appearance.
As I understand, the study is purely theoretical. The authors then need to give more information about the different programs used, not just mention them by name. I imagine that most of the readers are not very familiar to these programs.
Line 50
How can materials with lower atomic numbers have higher electron density?
Line 68
Please, define “payload”. Not all readers may be familiar to space technology vocabulary.
Table 1
It would be good to include also electron densities of the different materials.
Table 2
Motivate the choice of electron (2-7 MeV) and proton (25-200 MeV) energies that have been used in the calculations.
Fig. 2
What area is simulated? How are particles scattered sideways, or backwards treated? What is the distance between layers?
Line124
What is the need for the third layer (5 micron Al)? How much contribution to the dose in the detector comes from backscattering?
Line 226 (and onwards, Fig. 7)
As I understand, Fig 7 shows the remaining flux of 2 MeV electrons after penetrating the 3 mm A equivalent shielding. What was the flux before the shielding and how is this “particle flux calculation in MULASSIS” performed?
Regarding many of the plots, please put the origin (0,0) in the lower left corner. To me, it looks strange with negative axes in these cases.
Eq. 1
Explain all the letters of the equation (dx/dt looks like a derivative, i.e. velocity).
Eq. 2
Explain E0 and Emax.
Eq. 3
Mysub(en) and Rho should be explained.
Line 267-269
Please, give references to this statement.
Figs 12, 13
Change “Deep” to “Depth”.
Reviewer 2 Report
In the introduction, authors briefly describe shielding against radiation in space. Later in this section, they mention the SHIELDOSE-2 software, which is commonly used to calculate depth doses using aluminum shielding. Other materials can be "simulated" by comparing areal densities equivalent to aluminum. This may lead to systematic over- or underestimation of the doses if calculated for exact materials. For these calculations, the authors chose a code named MULASSIS, based on the GEANT4 libraries. I find this section informative and well-written.
In the next section, the authors provide some information about the geometry used for benchmark calculations for six materials (five plus aluminum as the reference - Table 1).
Remarks:
a) In this section, I miss technical details for the used MULASSIS code, which are crucial to cross-check the results presented in the paper. Firstly, is there any version number of the MULASSIS code? The reference is pointing to a paper from 2002. What exact GEANT4 version was used, and what was the physics list used in Geant4? All of this is missing; please add that kind of information for completeness.
b) Please provide references for all the densities used in Table 1.
In section three, calculation results are presented for various electron energies (2 to 7 MeV) and proton energies (25 to 200 MeV). Figure 3 shows significant differences in the doses in the silicon detector calculated for six shielding materials (with equivalent areal density).
Remarks:
c) Please note that percentage differences in Table 3 are calculated under the assumption that all of the electron energies contribute equally. Please write it explicitly. One can ask what should be a more realistic electron/or proton energy profile?
d) In the subsection dedicated to proton shielding, please provide differences in a similar table as Table 3 for electrons.
e) Figure 5 data is not easy to distinguish differences. Can you add (maybe as an inset of another figure) a plot of doses normalized to the dose after using aluminum shielding as a function of proton energy?
In the fourth section, the discussion, authors try to explain observed differences looking at calculated variables - Figures 7 to Figure 13.
Remarks:
f) What is missing here, and in the rest of the paper, is a comparison of the calculated observables by MULASSIS to experimentally measured ones, or providing a reference where energy deposits in the six materials by electrons and protons were investigated.
g) Please write about the possible errors related to simulations - systematic and statistical (if any).
In general, I would recommend the paper for publication as it attempts to increase the precision of shielding efficiency. However, only after a major revision that addresses the remarks stated above [a-g] with the most important are points a,f and g.
Round 2
Reviewer 2 Report
Thank you for the response, after corrections, I recommend the article to be published in the Electronics journal.
